# Towards Universal Mono-to-Binaural Speech Synthesis

## Abstract

We consider the problem of synthesis of binaural speech from mono audio in arbitrary environments, which is important for modern telepresence and extended-reality applications. We find that existing neural mono-to-binaural methods are overfit to non-spatial acoustic properties, via analysis using a new benchmark (TUT Mono-to-Binaural), the first introduced since the original dataset of Richard et al. (2021). While these past methods focus on learning neural geometric transforms of monaural audio, we propose BinauralZero, a strong initial baseline for *universal* mono-to-binaural synthesis, which can subjectively match or outperform existing state-of-the-art neural mono-to-binaural renderers trained in their target environment despite *never seeing any binaural data*. It leverages the surprising discovery that an off-the-shelf mono audio denoising model can competently enhance the initial binauralization given by simple parameter-free transforms. We perform comprehensive ablations to understand how BinauralZero bridges the representation gap between mono and binaural audio, and analyze how current mono-to-binaural automated metrics are decorrelated from human ratings.

## 1 Introduction

Humans possess a remarkable ability to localize sound sources and perceive the surrounding environment through auditory cues alone. This sensory ability, known as *spatial hearing*, plays a critical role in numerous everyday tasks, including identifying speakers in crowded conversations and navigating complex environments (Blauert, 1996). Hence, emulating a coherent sense of space via listening devices like headphones is key to creating truly immersive artificial experiences. The case of position-conditional binaural rendering of mono speech audio is of special interest, due to growing reliance on remote real-time spoken interactions in professional settings, increased prevalence of high-fidelity extended-reality (XR) technologies, and the socially cohesive benefits of spatial audio in virtual spaces (Lawrence et al., 2021; Lieberman et al., 2022; Nowak et al., 2023). In particular, these demands motivate speech spatialization schemes that are *universal*, accurately emulating the relative position of the source speaker, appropriately conditioned on (or performing a generic imputation of) room and binaural listener properties, all while being robust to the identity of the speaker, to the content and language of the speech, and mitigating ambient noise.

Conventional digital signal processing approaches often involve linear time-invariant (LTI) systems with explicit models for the head-related transfer function (HRTF), the room impulse response (RIR), and ambient noise (Savioja et al., 1999; Zotkin et al., 2004; Sunder et al., 2015; Zhang et al., 2017). To reduce explicit linearity assumptions and modeling choices, Richard et al. (2021) demonstrated that for mono-to-binaural speech synthesis, direct deep supervised learning outperforms such approaches on both loss-based and human evaluations on their introduced real-world dataset. Their choice of architecture and training scheme has been refined by a body of subsequent work (Huang et al., 2022; Leng et al., 2022; Lee & Lee, 2023; Liu et al., 2023; Kitamura & Itou, 2023; Li et al., 2024b).

However, we find that existing neural approaches significantly overfit to the non-spatial acoustic properties of their data, representing a large gap from achieving universal mono-to-binaural synthesis. Though overfitting is most directly resolved by large-scale data collection, supervised data involves positional tracking of mono audio sources plus a binaural recording device atop a real or emulated human torso. For example, the original two-hour dataset of Richard et al. (2021) has remained the only dataset used by these works (except for an unreleased set that Huang et al., 2022 additionally

use); it is recorded in a single non-anechoic room, with the same set of eight speakers in the train and test data. Hence, we propose an alternate approach to mono-to-binaural synthesis that our experiments suggest can scale to universal binaural rendering, or at least represents a strong environment-agnostic baseline towards it. In particular, we discover the "(mono audio, source position) ↦ binaural audio" task can be precomposed with parameter-free transforms into mono audio enhancement tasks that can be performed surprisingly well by off-the-shelf denoising audio models, such as those found in text-to-speech vocoders. Finally, we analyze our approach's design choices and the limitations of automated metrics across systems revealed by our work. Explicitly, our contributions include:

- Showing that existing neural models highly overfit to non-spatial acoustic features. This includes releasing the **first new benchmark dataset for the task (TUT Mono-to-Binaural)**, using ambisonic recordings of anechoic speech (TUT Sound Events 2018 ANSYN; Adavanne et al., 2019) that we reparameterize into binaural recordings, on which pretrained models degrade significantly.

- **BinauralZero**, a novel, **state-of-the-art baseline for *universal* neural mono-to-binaural audio synthesis**, leveraging parameter-free transforms (geometric time warping, amplitude scaling), and an off-the-shelf denoising vocoder (WaveFit; Koizumi et al., 2022a). Despite **training on zero binaural data, its syntheses are perceptually on-par or better than supervised methods** trained entirely Richard et al. (2021)'s dataset (similarity, spatialization, naturalness), while greatly outperforming them on our new TUT Mono-to-Binaural benchmark.

- Ablations to BinauralZero to **analyze how denoising and warping close the representational gap of mono audio and its binaural perception**. Based on the automated loss metrics attained by our training-free method versus existing work, we find that **current phase, amplitude, waveform, and STFT metrics can mislead when comparing in-domain** neural mono-to-binaural systems, and mathematically derive properties of these metrics in high-error regimes.

## 2 REVISITING MONO-TO-BINAURAL SYNTHESIS

### 2.1 BACKGROUND

The reproduction of virtual acoustic environments has been modeled as room- and listener-based transformations of directional source audio, as expressed as LTI systems in DSP via convolutional application of RIRs and HRTFs, respectively (Savioja et al., 1999). However, the computational cost of wave-based RIR simulation (Välimäki et al., 2012) and the collection cost of measuring HRTFs (Li & Peissig, 2020) lead to the use of simplified geometric models and generic HRTFs in practice (Sunder et al., 2015). Motivated by the difficulty of collecting HRTF and RIR data, Gebru et al. (2021) showed that an implicit HRTF can be learned by a temporal CNN, Richard et al. (2022) and Lee et al. (2022) showed that neural networks can estimate RIR filters from training data, and Luo et al. (2022) learn an implicit neural representation of an acoustic field for spatial audio generation. These works motivate using deep learning to supersede an explicit binaural reproduction pipeline. Hence, Richard et al. (2021) proposed one of first uses of neural networks for mono-to-binaural synthesis, composing a neural time-warping module (WarpNet) and a temporal (hyper-)convolutional neural network (CNN) to directly map mono audio to binaural waveforms. BinauralGrad (Leng et al., 2022) was the first to use a denoising diffusion probabilistic model (DDPM), composed of two stages: the first denoises a channel-averaged waveform, then the second conditions on this, the original mono audio, and their geometric warps to jointly denoise both channels.

Since then, better incorporation of the inductive biases from DSP have led to neural systems that are more efficient or improve objective rendering metrics. Neural Fourier Shift (NFS; Lee & Lee, 2023) predicts delays and scaling from speaker locations and match the above methods' perceptual spatial similarity with a much smaller model. Huang et al. (2022) show that mono-to-binaural audio synthesis can be combined with the use of discrete audio codes to improve spectral loss. Kitamura & Itou (2023) used a structured state space sequence (S4) model for the mono-to-binaural task and attain similar loss metrics to above works. To improve the phase loss of their chosen systems, DopplerBAS (Liu et al., 2023) incorporated the Doppler effect in the conditioning of WarpNet and BinauralGrad, and DIFFBAS (Li et al., 2024b) proposed an interaural phase difference loss atop WarpNet and NFS.

Finally, there is a broader body of work using different conditioning settings for multi-channel audio. One line of work uses visual conditioning for the generation of binaural audio (Xu et al., 2021; Parida et al., 2022; Chen et al., 2023a;b; Liang et al., 2023; Somayazulu et al., 2023; Garg et al., 2021;

Yoshida et al., 2023; Xu et al., 2023; Li et al., 2024c;d; Liu et al., 2024). Also, for music applications there is a generative task, where plausible and subjectively appealing binaural renderings are imputed from a single-channel recording of multi-source audio (e.g., Chun et al., 2015; Serrà et al., 2023; Li et al., 2024a; Zang et al., 2024; Zhu et al., 2024).

## 2.2    A New Benchmark: TUT Mono-to-Binaural

Given the ongoing use of Richard et al. (2021)'s baseline **Binaural Speech** dataset[1] by existing works despite its small training set (two hours) and fixed acoustic properties (room, language, shared bank of speakers in train and test, maximal distance of 1.5m), we set out to define a simple test-only benchmark to assess whether mono-to-binaural models trained on Binaural Speech and future datasets are retaining basic binaural rendering functionality in a relatively clean setting.

Hence we build **TUT Mono-to-Binaural**, a simple and analogous benchmark which we release at `[URL at camera-ready; see Supplementary Material for examples]`. It demonstrates a new approach to collecting task data by pairing reference mono audio with binaural projections from their multi-channel *ambisonic* recordings. We start from the overlap-free audio (OV1) in the TUT Sound Events 2018 ANSYN sound localization dataset[2] (Adavanne et al., 2019), which takes real monophonic recordings from the DCASE 2016 Task 2 dataset[3] and spatializes at varying elevations, azimuths, and distances into anechoic first-order Ambisonic (FOA) recordings, with four audio channels to cover 3D space; see Adavanne et al. (2018) for more details. Overall, there are around 2 hours of recordings in the dataset. In particular, the original monophonic recordings include spoken French sentences sampled at 44.1 kHz with an AT8035 shotgun microphone connected to a Zoom H4n recorder (Mesaros et al., 2018). We then convert the FOA's location annotations (elevation, azimuth, distance) into a Cartesian coordinate system $p^{\text{src}} = (x, y, z)$ to match the format of Binaural Speech. Next, ground-truth metadata was leveraged to cut out the speech segments from the FOA recordings using their provided timestamps. Finally, the FOA recordings are rendered as binaural audio using OmniTone,[4] a well-established commercial DSP ambisonic decoder that projects the highly spatial FOA recording down into a binaural rendering. This gives 1,174 binaural speech segments, each about 2s, corresponding to each's own 3D location. These become ground truths for the original DCASE 2016 Task 2 mono audio with their converted Cartesian coordinates.

The key idea is that TUT is acoustically and spatially simpler (anechoic room, stationary speech) while being out-of-domain in the speech itself (unseen speaker, unseen language, unseen microphone, broader elevation coverage, distances up to 10m) so if supervised models have learned to model and generalize spatial properties rather than acoustic confounders, we would expect them to still produce reasonable renderings after training only on Binaural Speech or future mono-to-binaural corpora.

## 2.3    Measuring Generalization via Human Evaluation

Prior work defines a number of automated and human evaluations to assess mono-to-binaural rendering. Later in this work we find that automated metrics decorrelate with perceptual metrics (Section 4.2), so for now we focus on the ultimate goal of matching the ground truth with regards to human spatial hearing, under the existing benchmark and our proposed benchmark. Following precedent from past work, for reference-free evaluations we use **mean opinion score (MOS)**. For reference-based evaluations we use the more sample efficient **multiple stimuli with hidden reference and anchor (MUSHRA)**, especially as references are generally canonical in binaural audio (unlike in text-to-speech). We categorize the human evaluations in literature into three broad axes:

- **Naturalness:** The overall naturalness and intelligibility of the synthesized audio content. We capture this as **naturalness MOS (N-MOS)**, which is analogous to (regular) MOS in Leng et al. (2022), or to cleanliness plus part of realism MOS in Richard et al. (2021).

---

[1] `https://github.com/facebookresearch/BinauralSpeechSynthesis/releases/tag/v1.0`

[2] `https://zenodo.org/records/1237703`

[3] `https://archive.org/details/dcase2016_task2_train_dev`

[4] `https://googlechrome.github.io/omnitone/#home`

- **Spatialization:** How realistic the synthesis is as a rendering of binaural audio. We capture this as **spatialization MOS (S-MOS)**, which is analogous to spatialization MOS in Leng et al. (2022), or to spatialization MOS plus part of realism MOS in Richard et al. (2021).

- **Similarity:** How similar the synthesized audio is to the reference spatial audio. We capture this as **(similarity) MUSHRA**, which is the MUSHRA analogue to the reference-provided similarity MOS in Leng et al. (2022) and a generalization of spatial MUSHRA as in Lee & Lee (2023).

We consider the three primary neural models (WarpNet, BinauralGrad, NFS), each of which released their pretrained Binaural Speech models. We take these models adapted on the Binaural Speech dataset and test them on Binaural Speech and our new proposed TUT Mono-to-Binaural benchmark. Finally, we include our proposed BinauralZero (Section 3), which has not seen either Binaural Speech or TUT Mono-to-Binaural (or any binaural data at all). See Appendix B for formal evaluation and implementation details. Our MOS results are in Table 1 and our MUSHRA results are in Figure 1:

Table 1: Reference-free human evaluations (naturalness and spatialization MOS) of neural methods.

| Type | Model | Binaural Speech | | TUT Mono-to-Binaural | |
| --- | --- | --- | --- | --- | --- |
| | | N-MOS (↑) | S-MOS (↑) | N-MOS (↑) | S-MOS (↑) |
| Supervised | WarpNet | 3.86±0.16 | 3.73±0.27 | 3.60±0.26 | 2.99±0.22 |
| (on Binaural | BinauralGrad | 4.01±0.14 | 3.56±0.23 | 3.27±0.32 | 2.29±0.23 |
| Speech) | NFS | 3.99±0.15 | 3.53±0.22 | 3.79±0.23 | 2.89±0.26 |
| Unadapted | BinauralZero (ours) | 4.07±0.17 | 3.76±0.25 | 3.98±0.15 | 3.73±0.21 |
| Ground Truth | | 4.30±0.12 | 3.99±0.20 | 4.08±0.11 | 4.03±0.26 |

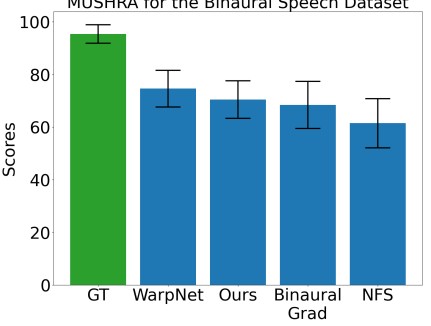 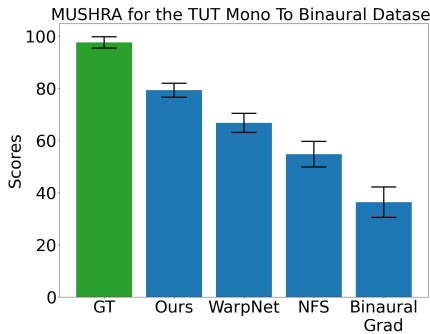

Figure 1: MUSHRA scores for the Binaural Speech dataset and our TUT Mono-to-Binaural benchmark. Higher is better, with the upper bound determined by how the hidden reference was scored (labeled GT, which should be close to 100). The specific numerical values are reported in Appendix B.

We see that models can fail to generalize within each axis. For example, we see that on the new evaluation set, WarpNet and NFS remain generally performant on naturalness (considering the ground truth's N-MOS has also decreased) but degrade significantly on spatialization and partly on similarity. Upon inspection, one hears two respective failure modes: (a) incorrect spatialization, manifesting as generated binaural speech with unrealistic distance cues or spatial artifacts when beyond the training range, and (b) dissimilarity from not retaining the original speaker's voice characteristics in the binaural output. We also see that despite having the highest score versus the other supervised methods in spatiality MOS, and equal-to-highest naturalness MOS, NFS's MUSHRA score is notably lower than all other neural methods; reflecting their focus on parameter efficiency and the strong inductive bias of rendering in Fourier space, which favors spatial performance and generalization but leaves less capacity for e.g., speaker invariance. Furthermore, BinauralGrad degraded on all metrics, producing outputs with substantial Gaussian noise, suggesting the diffusion process does not generalize outside the specific acoustics of the training distribution. These failures can be heard in the binaural rendering examples at [URL at camera-ready; see Supplementary Material for examples].

Though these three axes are entangled, our results make the case that **future work in mono-to-binaural synthesis should have a 'basis' of evaluations spanning all three aspects**. We note that

only Leng et al. (2022) covered all three axes in human evaluation; Richard et al. (2021) covers the first two, Huang et al. (2022) and Lee & Lee (2023) focus on spatialization similarity, and Liu et al. (2023); Kitamura & Itou (2023); Li et al. (2024b) do not perform human evaluations.

Stepping back, we see **that models adapted to a room- and speaker-specific dataset like Binaural Speech regress in perceptual naturalness, spatialization, and ground-truth similarity on even the anechoic, stationary setup of TUT Mono-to-Binaural**, suggesting these deep neural networks of <10M parameters (Lee & Lee, 2023) are already not learning the appropriate features on these small datasets. In contrast, our proposed BinauralZero (described next section), is perceptually on-par or outperforms binaurally supervised methods on all three axis, despite not having seen any binaural data, suggesting an alternate path towards 'universal' mono-to-binaural speech synthesis.

## 3 BINAURALZERO: TOWARDS UNIVERSAL MONO-TO-BINAURAL SYNTHESIS

### 3.1 MOTIVATION

As discussed in Section 1 and 2.2, it is difficult to collect real-world data, especially over the universe of possible positions, source audio types, and acoustic conditions, to directly train strong supervised models that generalize past the two-hour training set. We also note that other mitigations like synthetic data generation, in-context prompting, or parameter-efficient finetuning exist; however, to our knowledge there are no strong multi-channel and/or spatially-aware audio models to facilitate quality pseudolabeling or a finetuning that does not involve learning representations for part of the input/output space from scratch. We leave such approaches to future work.

For now, we note there are strong monophonic self-supervised speech models trained on large data. A large class of these are denoising (diffusion) vocoders, which are able to output denoised waveforms conditional on some semantic information (e.g., speech tokens). We also know that denoising diffusion models are promising as an architecture, given BinauralGrad's success on training two position-conditional denoising diffusion models (though on requiring joint denoising of both channels) to outperform WarpNet on Binaural Speech despite similar parameter counts (6.9M vs. 8.6M); the main downside of which was having to train only on the two-hour Binaural Speech dataset, where its better in-distribution fitting made it more brittle out-of-domain.

The gap between using existing speech mono denoising models is (1) they only operate on individual channels, and (2) they are not trained to explicitly condition on position. However, we argue that (1) is not an inherent issue, as there is no strict reason to do multi-channel rendering jointly as in BinauralGrad (other than for parameter efficiency / regularization) if enough conditioning information is given; recall binaural hearing is observing the same underlying soundscape from two ear positions.

As for (2), we note that denoising waveform models learn to denoise at varying noise levels, which means that we can implicitly perform conditioning by providing an almost-complete waveform. The vocoder does not even have to be trained on content-diverse data, as the behavior we need is cleaning up direction-related artifacts, which should vary (along with distances and recording conditions) if pretrained on a large corpus. It is then plausible that the 'denoising basin' of a such a vocoder is able to fix slight issues in a hypothesized spatial transformation. In this work we consider **geometric time-warping**, whose parameter-free version was used in WarpNet and subsequent works; and **amplitude scaling**, which we are the first to explicitly apply to neural mono-to-binaural synthesis.

Remarkably, this overall scheme requires zero binaural data, and thus we name it **BinauralZero**. It is summarized in Figure 2; the algorithm is also formally described in Appendix D as Algorithm 1. Note that our method does not take into account room effects nor the listener's head shape. Thus, one interpretation is that BinauralZero produces spatial audio which imputes both a generalized low RIR room (regularized by all the data the vocoder was trained on) and an implicit generic HRTF.

### 3.2 GEOMETRIC TIME WARPING (GTW)

GTW estimates a warpfield that separates the left and right binaural signals by applying the interaural time delay (ITD) based on the relative positions of the sound source and the listener's ears. Richard et al. (2021) proposed GTW as a method to generate an initial estimate of the perceived signals. This approach offers a simple and parameter-free solution for warpfield which can be applied to the mono

Figure 2: Our proposed BinauralZero method, our state-of-the-art training-free baseline for universal mono-to-binaural speech synthesis. Mono waveform is binauralized with geometric time warping, conditional on the speaker's position, then the two channels' amplitudes are scaled to prime interaural level differences. Each channel is then denoised $N = 3$ times by a low-noise-level step of a (mono) denoising spectrogram-conditional text-to-speech vocoder.

signal. Let $S$ denote the signal's sample rate and $\nu_{\text{sound}}$ represent the speed of sound. The system employs basic GTW on the monaural signal $x$. This warping is achieved by computing a warpfield for both the left and right listening channels, denoted by $\rho^\ell(t), \rho^r(t)$. The values of this warpfield are computed using on the source and listener ear positions $\boldsymbol{p}_t^{\text{src}}, \boldsymbol{p}_t^\ell, \boldsymbol{p}_t^r$:

$$\rho^\ell(t) := t - \frac{S}{\nu_{\text{sound}}} \, ||\boldsymbol{p}_t^{\text{src}} - \boldsymbol{p}_t^\ell||_2, \qquad \rho^r(t) := t - \frac{S}{\nu_{\text{sound}}} \, ||\boldsymbol{p}_t^{\text{src}} - \boldsymbol{p}_t^r||_2 \qquad (1)$$

As this function takes non-integer values, we can define the warped left and right signals $\hat{x}^\ell, \hat{x}^r$ with respect to the original indexing $t$ via linear interpolation:

$$x_t^\ell := (\lceil \rho^\ell(t) \rceil - \rho^\ell(t)) \cdot x_{\lfloor \rho^\ell(t) \rfloor} + (\rho^\ell(t) - \lfloor \rho^\ell(t) \rfloor) \cdot x_{\lceil \rho^\ell(t) \rceil},$$
$$x_t^r := (\lceil \rho^r(t) \rceil - \rho^r(t)) \cdot x_{\lfloor \rho^r(t) \rfloor} + (\rho^r(t) - \lfloor \rho^r(t) \rfloor) \cdot x_{\lceil \rho^r(t) \rceil}.$$

### 3.3 AMPLITUDE SCALING (AS)

In addition to manipulating the time-delay of the signal, we also manipulate the amplitude of the signal based on the position of the speaker. Human spatial perception of sound relies on various factors, including the ITD, the interaural level difference (ILD), and spectral cues due to HRTFs. While prior works (Wersényi, 2010; Baumgarte & Faller, 2003) suggest that the ILD is mostly caused by scattering off of the head and is dominant in human spatial perception for sounds with high frequencies, we find that amplitude scaling based on the inverse square law has a positive effect on the perceived spatial accuracy of the processed signal.

Our approach aims to leverage this amplitude manipulation to enhance the spatial realism of the generated binaural audio. Let $D$ be the Euclidean distance from the origin of the sound waves. Then by the inverse-square law, pressure drops at a $1/D^2$ ratio (Zahorik et al., 2005). In the case of microphones, pressure manifests as amplitude. Acknowledging that the left-right microphone distance of the KEMAR mannequin used in datasets like Richard et al. (2021)'s is only an approximation of human heads, we define:

$$D_t^\ell = ||\boldsymbol{p}^{\text{src}} - \boldsymbol{p}_t^\ell||_2, \qquad D_t^r = ||\boldsymbol{p}^{\text{src}} - \boldsymbol{p}_t^r||_2. \qquad (2)$$

Then, at each time step we scale down the magnitude of the side furthest from the source, using the ratio of the closer side's distance versus the further side's distance:

$$\hat{x}_t^\ell := \min(1, (D_t^r/D_t^\ell)^2) \cdot x_t^\ell, \qquad \hat{x}_t^r := \min(1, (D_t^\ell/D_t^r)^2) \cdot x_t^r. \qquad (3)$$

### 3.4 DENOISING VOCODER

GTW and AS are simple, parameter-free operations that only roughly approximate binaural audio; using the warped and scaled speech signals $\hat{x}^\ell, \hat{x}^r$ as-is results in acoustic artifacts and inconsistencies.

Hence, there is a need for further refinement to generate natural-sounding binaural audio. To this end, we propose that a sufficiently well-trained denoising vocoder could be used on each signal *independently*. We use a WaveFit neural vocoder (Koizumi et al., 2022a) as our denoising vocoder model. It is a fixed-point iteration vocoder that takes the denoising perspective of DDPMs (Ho et al., 2020); and takes the discriminator of generative adversarial networks, specifically MelGAN's (Kumar et al., 2019), to learn a sampling-free iterable map that can generate natural speech from a degraded input speech signal. As a vocoder, it takes log-mel spectrogram features and noise as input and produces clean waveform output. In WaveFit's notation, we perform the iterated application of

$$\hat{y}_{i-1} := \mathcal{V}_\theta(\hat{y}_i, \boldsymbol{c}, k) := \mathcal{G}(\hat{y}_i - \mathcal{F}_\theta(\hat{y}_i, \boldsymbol{c}, k), \boldsymbol{c}), \qquad (4)$$

where $\boldsymbol{c}$ is the spectrogram to convert, $\hat{y}_{i-1}$ is a candidate waveform refined from $\hat{y}_i$, and $k$ is the time-step. $\mathcal{G}$ is a parameter-free gain adjustment operator and $\mathcal{F}_\theta$ is the WaveGrad architecture (Chen et al., 2021) trained for reconstruction under a discriminator.

WaveFit is pretrained such that the starting noise is given by $\hat{y}_K \sim \mathcal{N}(0, \boldsymbol{\Sigma_c})$ where $\boldsymbol{\Sigma_c}$ is a covariance matrix initialized as in SpecGrad (Koizumi et al., 2022b) to capture the spectral envelope of $\boldsymbol{c}$; both $k, i$ iterate over $K, \ldots, 1$. However, for BinauralZero, we express our "approximation" hypothesis by iterating at the noise level of WaveFit's *final* denoising step ($k = 1$). We then iteratively denoise $\hat{y}_N^\ell, \hat{y}_N^r := \hat{x}^\ell, \hat{x}^r$, conditioning on their initial log-mel spectrograms and the fixed low noise level for steps $i = N, \ldots, 1$.

## 4 RESULTS AND DISCUSSION

We use a WaveFit vocoder as described in Koizumi et al. (2022a), pretrained on the 60k-hour LibriLight dataset, which is an untranscribed corpus of open-source English audiobooks derived from the LibriVox project (Kahn et al., 2020). The pretraining hyperparameters used are as in Koizumi et al. (2022a), giving 13.8M parameters.

### 4.1 HUMAN EVALUATIONS AND THEIR LIMITATIONS

We reported the human evaluation results of BinauralZero in Section 2.3 (Table 1 and Figure 1), but now discuss them here. On Binaural Speech (which, unlike BinauralZero, all supervised methods were trained on), our subjective evaluation results show that BinauralZero improves in N-MOS over WarpNet, BinauralGrad and NFS by 0.21, 0.06 and 0.08, while attaining similar S-MOS. MUSHRA results (Figure 1) show that human raters do not have a statistically significant preference for any of the methods WarpNet, BinauralGrad, NFS or BinauralZero, similar to the spatial-specific MUSHRA conclusions of Lee & Lee (2023).

On the simpler TUT Mono-to-Binaural dataset however, we see that BinauralZero is the only one to maintain performance, whereas all other methods sharply degrade. For example, BinauralZero maintains an average S-MOS of above 3.7, whereas other systems degrade to an average S-MOS of 3.0 or less. The smaller and disjoint error bars on MUSHRA for TUT Mono-to-Binaural (Section 2.3) show their performances on it are easily distinguishable, with BinauralZero outperforming other mono-to-binaural methods in a significant way and performing close to the ground truth.

Samples can be heard at [URL at camera-ready; see Supplementary Material for now]. Note that as BinauralZero does not condition on room information (in particular, ours uses a vocoder derived from studio audiobook recordings), its syntheses can lack distance or reverb versus the ground truth, which may be underrated in a generic 'similarity' prompt. Future universal-type approaches that optionally condition on room information should consider finer similarity tasks focusing on closeness in position like in Huang et al. (2022), or coherence over different-positioned renderings.

### 4.2 AUTOMATED EVALUATIONS AND THEIR LIMITATIONS

For reference-based automated evaluations, we consider the same objective metrics as in prior work:

- **Wave $\boldsymbol{\ell_2}$**: mean squared error (MSE) between the ground truth and synthesized per-channel waveforms. This metric is multiplied by $10^3$.

- **Amplitude $\ell_2$**: MSE between the STFTs of the ground truth and synthesized audio, with respect to amplitude.

- **Phase $\ell_2$**: MSE between the left-right phase angle of the ground truth and synthesized audio. Phase is computed from the STFT.

- **Multi-resolution STFT ($\mathcal{L}_{\mathbf{STFT}}$)** is the multi-resolution spectral loss on STFTs.

Unlike previous work, we do not report PESQ scores. Lee & Lee (2023) already found that large deviations here (1.66 vs. 2.36, 2.76) did not indicate a significant difference in subjective spatial similarity; furthermore, our investigation of open source code from previous work shows that these were computed only on the left channel of the audio input. As with the human evaluations, we evaluate on both the Binaural Speech test set as well as TUT Mono-to-Binaural. We also include a DSP baseline on Binaural Speech; we use the open-source Resonance Audio package,[5] which takes speaker and listener locations, room size, and room materials as input. For each dataset, room size is configured base on dataset definition and room materials are configured based on standard building materials; exact configurations are presented in Appendix C. Our results are in Table 2 and Table 3, with the (reference-based) MUSHRA human evaluations included for reference.

Table 2: Reference-based automated metrics of models on the Binaural Speech test set. Similarity MUSHRA scores are included for reference.

| TYPE | MODEL | WAVE $\ell_2$ ($\downarrow$) | AMP $\ell_2$ ($\downarrow$) | PHASE $\ell_2$ ($\downarrow$) | $\mathcal{L}_{\mathrm{STFT}}$ ($\downarrow$) | MUSHRA ($\uparrow$) |
|------|-------|------|------|------|------|------|
| ADAPTED | DSP (OURS) | 0.812 | 0.052 | 1.572 | 1.91 | – |
| | WARPNET | 0.179 | 0.037 | 0.968 | 1.52 | 74.6±7.0 |
| | BINAURALGRAD | 0.128 | 0.030 | 0.837 | 1.25 | 68.4±9.0 |
| | NFS | 0.172 | 0.035 | 0.999 | 1.29 | 61.5±9.4 |
| UNADAPTED | BINAURALZERO (OURS) | 0.440 | 0.053 | 1.508 | 1.91 | 70.5±7.1 |

In Table 2, we observe that BinauralZero achieves significant objective improvements over the DSP baseline, despite not modeling additional interactions between the two generated channel streams or the RIR and HRTF. However, BinauralZero underperforms the supervised neural methods in *all* reference-based automated metrics. In terms of Wave $\ell_2$, BinauralZero underperforms the supervised methods WarpNet, BinauralGrad and NFS, with a 2-3x larger loss. On the remaining losses, BinauralZero has a loss that is at least 25% above the next method's. Despite uniformly worse automated metrics, the perceptual similarity performance of BinauralZero method is at least comparable to the other methods (if not better, e.g. versus NFS), even though BinauralZero has not been trained on the Binaural Speech dataset. This does not even account for the better reference-free N-MOS and comparable-to-better S-MOSes (Section 4.1), approaching that of the ground truth.

The Phase $\ell_2$ is also close to $\pi/2$ for BinauralZero and DSP on Binaural Speech, which suggests a high-error regime in a numerical sense (see Lemma 1 below). However, despite supervised models attain $\leq 1$ in Phase $\ell_2$, this **reduction in phase loss does not lead to measurable *perceptual* gains** over BinauralZero, even during explicit side-by-side evaluation via similarity MUSHRA. This is notable as Richard et al. (2021) speculated on the importance of phase estimation in binaural audio due to human sensitivity to ITDs as small as $10\mu s$ (Brown & Duda, 1998), leading to existing works' addition of a phase term to the objective to induce this; however, they did not specifically ablate their loss modification in human evaluations. In contrast, text-to-speech vocoders like WaveFit design their loss functions to avoid such imperceptible improvements (see Section 4.2 of Koizumi et al., 2022a). Our results show that, surprisingly, **the failure of off-the-shelf mono vocoders to model phase is not a notable issue** for their use in channelwise binaural denoising. Future work could remedy this by devising a phase-aware adaptation scheme for BinauralZero on binaural speech.

These results suggest that **all current automated metrics in neural mono-to-binaural speech synthesis are uninformative when in-domain**. Notably, we find their uninformativeness happens well before the loss values attained by the original baseline of WarpNet (Richard et al., 2021) which first reported these metrics. They could even be *misleading*; for example, NFS outperforms

---

[5] https://github.com/resonance-audio

WarpNet on three of four objective metrics but is significantly worse than WarpNet on similarity MUSHRA (61.5 vs. 74.6). This also qualifies results like Liu et al. (2023); Kitamura & Itou (2023); Li et al. (2024b), which drop human metrics; it remains unclear whether their improvements are perceptible versus entirely due to improved fitting of imperceptible environment-specific artifacts, like high-frequency recording equipment noise.

Table 3: Reference-based automated metrics of models on the TUT Mono-to-Binaural benchmark. Similarity MUSHRA scores are included for reference.

| TYPE | MODEL | WAVE $\ell_2$ ($\downarrow$) | AMP $\ell_2$ ($\downarrow$) | PHASE $\ell_2$ ($\downarrow$) | $\mathcal{L}_{\text{STFT}}$ ($\downarrow$) | MUSHRA ($\uparrow$) |
|---|---|---|---|---|---|---|
| ADAPTED | DSP (OURS) | 1.134 | 0.075 | 1.572 | 2.93 | – |
| (TO BINAURAL | WARPNET | 2.909 | 0.099 | 1.571 | 6.66 | 66.7±3.6 |
| SPEECH) | BINAURALGRAD | 3.228 | 0.218 | 1.571 | 5.40 | 36.4±5.8 |
| | NFS | 1.574 | 0.085 | 1.571 | 3.06 | 54.7±4.9 |
| UNADAPTED | BINAURALZERO (OURS) | 0.293 | 0.045 | 1.572 | 2.93 | 79.3±2.7 |

In Table 3, we see that on our proposed anechoic, stationary TUT Mono-to-Binaural benchmark, BinauralZero significantly outperforms all methods that were adapted towards Binaural Speech, in both automated and perceptual metrics. Complementary to the previous observation, we see that the systems that are perceptually distinguishable have far larger metric differences than anticipated in previous work; e.g., WarpNet has 10x the Wave $\ell_2$ loss of BinauralZero to give a 12.6 (out of 100) absolute difference in MUSHRA. We also see that the Binaural Speech DSP baseline outperforms all Binaural Speech neural baselines on TUT Mono-to-Binaural, suggesting that existing neural adaptation schemes may come with a direct tradeoff away from handling TUT Mono-to-Binaural's baseline setting, making the current low-resource situation not tenable for achieving universal mono-to-binaural speech synthesis and hence motivating approaches like BinauralZero.

That said, we make the caveat that understanding automated evaluations can still aid model development, by deriving a relationship between phase + amplitude errors and the *relative* frequency-domain distance, when the latter is large–a numerical "high-error" regime. Adopting the notation from Richard et al. (2021), let $Y$ represent the audio signal in the frequency domain, and $\hat{Y}$ a model's prediction, with $\varepsilon$ denoting the distance between them. Our analysis distinguishes between high- and low-error regimes, defined by $\varepsilon/|\hat{Y}| \gg 1$ and $\varepsilon/|\hat{Y}| \ll 1$, respectively. For high error:

**Lemma 1.** *Let $\hat{Y} \in \mathbb{C}$, and let there be a sphere of complex numbers with distance $\varepsilon$ from $\hat{Y}$ such that $Y \in \mathbb{S}_\varepsilon = \{Y \in \mathbb{C} : |Y - \hat{Y}| = \varepsilon\}$. Assuming a high (relative) error regime $\frac{\varepsilon}{|\hat{Y}|} \gg 1$, the expected phase and amplitude error can be expressed as:*

$$(a) \quad \mathbb{E}_Y \left( \mathcal{L}^{(phase)}(Y, \hat{Y}) \right) \approx \frac{\pi}{2}, \qquad (b) \quad \mathbb{E}_Y \left( \mathcal{L}^{(amp)}(Y, \hat{Y}) \right) \approx \varepsilon. \tag{5}$$

*Proof.* This follows from Lemma 1 of Richard et al. (2021) combined with first-order approximations induced by large error; see Appendix E for derivations. $\square$

Figure 3 qualitatively shows that Lemma 1 holds, and empirically we see that in Table 3 all models attain this $\pi/2$, consistent with them being unadapted or adapted away towards Binaural Speech's e.g. more constrained set of elevations. In Appendix F we give a complementary lemma for low error.

## 4.3 ABLATION STUDY OF BINAURALZERO

The significance of each core component within the proposed method (GTW, AS, and WaveFit) is evaluated through ablation studies (Table 4). All three components demonstrably contribute to the system's overall success. First, AS is critical for BinauralZero performance. Its absence leads to substantial degradation in both N-MOS and Wave $\ell_2$ error. Amplitude scaling between left and right channels creates a crucial perceptual difference, essential for accurate binaural audio modeling. GTW is the second most important component. Without GTW, left-right channel time differences become misaligned, resulting in increased Wave-$\ell_2$ error and decreased MOS. Interestingly, removing both

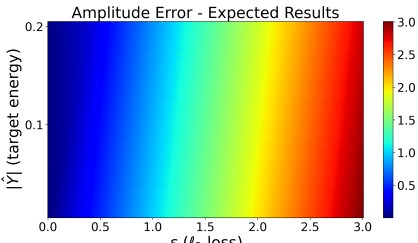 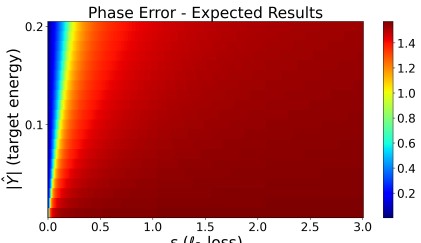

Figure 3: Expected errors from Richard et al. (2021) for reference, for amplitude and phase. We see that in bottom-right regions (the high-error regimes), the error magnitudes (represented by color) match our Lemma 1, being approximately $\epsilon$ or the fixed value $\pi/2$, respectively.

Table 4: Ablation of our BinauralZero method on the Binaural Speech dataset.

| MODEL | WAVE $\ell_2$ ($\downarrow$) | AMPLITUDE $\ell_2$ ($\downarrow$) | PHASE $\ell_2$ ($\downarrow$) | N-MOS ($\uparrow$) |
|---|---|---|---|---|
| BINAURALZERO | 0.440 | 0.053 | 1.508 | 4.07±0.17 |
| W/O AS | 0.802 | 0.059 | 1.539 | 2.93±0.16 |
| W/O GTW | 0.627 | 0.053 | 1.569 | 3.64±0.15 |
| W/O GTW, AS | 0.816 | 0.051 | 1.567 | 4.13±0.18 |
| DECODE FROM NOISE | 0.495 | 0.065 | 1.534 | 2.50±0.16 |
| W/O DENOISING (WAVEFIT) | 0.539 | 0.044 | 1.572 | 3.52±0.16 |
| DENOISING $\rightarrow$ AS $\rightarrow$ GTW | 0.474 | 0.072 | 1.277 | 3.85±0.19 |
| GTW $\rightarrow$ DENOISING $\rightarrow$ AS | 0.441 | 0.055 | 1.497 | 3.25±0.25 |
| 1 ITERATION | 0.459 | 0.069 | 1.393 | 3.62±0.20 |
| 2 ITERATIONS | 0.450 | 0.061 | 1.492 | 3.83±0.24 |
| 4 ITERATIONS | 0.445 | 0.053 | 1.502 | 3.94±0.18 |
| 5 ITERATIONS | 0.449 | 0.053 | 1.494 | 4.05±0.15 |

AS and GTW while retaining WaveFit leads to improved N-MOS, albeit resulting in a monaural waveform played identically in both channels (hence the degraded reference-based metrics).

In addition, we tested the effects of architectural modifications within the WaveFit inference process. Initializing with Gaussian noise (rather than the differentiated transformed waveforms) and decoding for five iterations, as in the original WaveFit implementation, resulted in poor audio quality. This is because the two channels remain independent, and playing them as a binaural recording produces an unaligned and noisy output. Also, any modification that does not conclude with denoising also degrades N-MOS, highlighting the importance of generating a natural self-consistent waveform. When removed in isolation, there is minimal impact on objective metrics but notable degradation. Applying WaveFit to the mono input first, followed by AS and GTW, yielded improved performance in terms of Phase $\ell_2$ but compromised Amplitude $\ell_2$ and N-MOS metrics. Likewise, applying AS at the end degraded N-MOS. Finally, increasing the number of denoising steps improves the objective metrics Wave $\ell_2$, Amplitude $\ell_2$ and Phase $\ell_2$ and improves N-MOS, but only until $N = 3$ iterations.

## 5 CONCLUSION

We considered the problem of position-conditional synthesis of binaural speech from mono audio across environments, which we term universal mono-to-binaural synthesis. We find that existing supervised learning schemes lose generalization ability due the low-to-zero resource nature of the task, by introducing a novel dataset specifically designed to test basic generalization ability of mono-to-binaural synthesizers. To motivate progress, we also described BinauralZero, a strong room- and listener-agnostic baseline that is generally performant. A universal model that can optionally condition on room and listener specifications is the clear next step, as well as improved automated metrics and finer-grained evaluations of coherence across syntheses in the same environment. Finally, we also made various empirical and theoretical recommendations of relevance to practitioners and system evaluators. Limitations and impacts are further discussed in Appendix A.

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

## A  LIMITATIONS AND BROADER IMPACT

BinauralZero uses an off-the-shelf neural vocoder which is conditioned on (log-mel) spectrogram features and no positional information which makes it difficult to condition towards a target phase spectrum. For this reason, our method struggles to directly and accurately process the phase information in binaural audio, leading to high Phase $\ell_2$ error. Furthermore, our method does not encode or use any room or head shape information. We hypothesize that this fact helps our method be competitive across room and acoustic environments, but fundamentally limits it from always matching supervised methods trained on a specific room and acoustic environment. Future work could learn to optionally

condition on such information. Future work could also consider non-speech sources, though we speculate that using a large-scale general-domain vocoder that has seen speech, music, and sound events may be sufficient to progress towards universal mono-to-binaural *audio* synthesis.

The proposed method employs a novel approach for enhancing mono audio signals into binaural audio. This technique has the potential to significantly improve the audio experience in augmented reality (AR) and virtual reality (VR) applications by creating a more immersive and realistic soundscape. The enhanced spatial audio cues generated by the proposed method can contribute to a heightened sense of presence and immersion within virtual environments. Additionally, the proposed method for transforming mono audio to binaural audio carries the potential for misuse in audio deepfake applications, where it could be employed to enhance the perceived realism and naturalness of manipulated audio through the introduction of artificially generated spatial cues.

## B  HUMAN EVALUATION DETAILS

For MOS, we collect mean opinion scores towards axes of naturalness. Human evaluators are tasked with assigning a rating on a five-point scale to denote the perceived naturalness of a given speech utterance, spanning from 1 (indicative of poor quality) to 5 (indicative of excellent quality). For every experiment, we use 50 random samples from each method. Every example is rated 5 times by different raters, with each experiment participated in by at least 30 raters. In the MUSHRA (multiple stimuli with hidden reference and anchor) evaluation, each question first presents the binaural recordings from the test set as a reference. The human raters are asked to rate how similar each model output is to the reference on a scale from 0 to 100. The samples include a hidden reference as an anchor, and the outputs of the models appear in random permutation order. For this test we used 50 random samples from each method. Following the MUSHRA protocol[6], we discard raters who gave >15% of hidden references a score below 90. We used the model and code releases of WarpNet[7], BinauralGrad[8], and NFS[9] to synthesize audio for subjective evaluations of these systems.

Table 5: MUSHRA results for the Binaural Speech dataset.

| SETTING | MODEL | MUSHRA (↑) |
|---|---|---|
| ADAPTED | WARPNET | 74.57±7.01 |
| | BINAURALGRAD | 68.40±8.99 |
| | NFS | 61.47±9.36 |
| UNADAPTED | BINAURALZERO (OURS) | 70.46±7.14 |
| GROUND TRUTH | | 95.37±3.53 |

Table 6: MUSHRA results for the TUT Mono-to-Binaural dataset.

| TYPE | MODEL | MUSHRA (↑) |
|---|---|---|
| ADAPTED | WARPNET | 66.71±3.61 |
| (TO BINAURAL | BINAURALGRAD | 36.35±5.84 |
| SPEECH) | NFS | 54.73±4.88 |
| UNADAPTED | BINAURALZERO (OURS) | 79.25±2.69 |
| GROUND TRUTH | | 97.59±2.19 |

[6]https://www.itu.int/dms_pubrec/itu-r/rec/bs/R-REC-BS.1534-3-201510-I!!PDF-E.pdf

[7]https://github.com/facebookresearch/BinauralSpeechSynthesis

[8]https://github.com/microsoft/NeuralSpeech/tree/master/BinauralGrad

[9]https://github.com/jin-woo-lee/nfs-binaural

## C  DSP CONFIGURATION

For room materials of both datasets, we used the configuration where left, right, front and back walls of the room are "brick-painted". For the down configuration (floor) we used the "curtain-heavy" configuration which simulates a rug. For the up (ceiling) configuration we used "acoustic-ceiling-tiles", as these are common in most office rooms and recording environments. As for room sized, for the binaural speech dataset, since it was recorded in a smaller room with a maximal distance of 1.5 meters from the microphone, we used a room configuration of width 4, height 3.5 and depth 4. For the TUT-mono-to-binaural dataset, since the maximal distance is 10 meters, we used a larger room with dimensions of width 12, height 3.5 and depth 12.

## D  ALGORITHM DEFINITION

---

**Algorithm 1** BinauralZero, our zero-shot mono-to-binaural algorithm:

---

**Require:** Denoising vocoder $\mathcal{V}_\theta$, iteration count $N$, low noise level $k$, and the following temporal sequences: mono waveform $x$, speaker position $\boldsymbol{p}^{\text{src}}$, listener's ear locations $\boldsymbol{p}^\ell, \boldsymbol{p}^r$.

$\quad x^\ell, x^r = \text{GeometricTimeWarping}(x, \boldsymbol{p}^{\text{src}}, \boldsymbol{p}^\ell, \boldsymbol{p}^r)$

$\quad \hat{x}^\ell, \hat{x}^r = \text{AmplitudeScaling}(x^\ell, x^r, \boldsymbol{p}^{\text{src}}, \boldsymbol{p}^\ell, \boldsymbol{p}^r)$

$\quad \boldsymbol{c}^\ell, \boldsymbol{c}^r = \text{LogMel}(\hat{x}^\ell), \text{LogMel}(\hat{x}^r)$

$\quad \hat{y}_N^\ell, \hat{y}_N^r := \hat{x}^\ell, \hat{x}^r$

$\quad \textbf{for } i \leftarrow N \text{ to } 1 \textbf{ do}$

$\quad\quad \hat{y}_{i-1}^\ell, \hat{y}_{i-1}^r = \mathcal{V}_\theta(\hat{y}_i^r, \boldsymbol{c}^r, k), \ \mathcal{V}_\theta(\hat{y}_i^r, \boldsymbol{c}^r, k)$

$\quad \textbf{end for}$

$\quad \textbf{return } \hat{y}^\ell, \hat{y}^r := \hat{y}_0^\ell, \hat{y}_0^r.$

---

# E    DERIVATIONS FOR LEMMA 1

## E.1    PHASE ERROR:

Utilizing the definition of the phase error as presented Lemma 1 of (Richard et al., 2021):

$$\mathbb{E}_Y\left(\mathcal{L}^{(\text{phase})}(Y,\hat{Y})\right) = \frac{1}{2\pi}\int_{-\pi}^{\pi}\arccos\frac{\text{Re}\left(\frac{\varepsilon}{|\hat{Y}|}\cdot e^{i\varphi}+1\right)}{\left|\frac{\varepsilon}{|\hat{Y}|}+e^{i\varphi}\right|}d\varphi \tag{6}$$

The integral over the phase $\varphi$ can be evaluated by the following steps:

$$\mathbb{E}_Y\left(\mathcal{L}^{(\text{phase})}(Y,\hat{Y})\right) = \tag{7}$$

$$= \frac{1}{2\pi}\int_{-\pi}^{\pi}\arccos\frac{\text{Re}\left(\frac{\varepsilon}{|\hat{Y}|}\cdot e^{i\varphi}+1\right)}{\left|\frac{\varepsilon}{|\hat{Y}|}+e^{i\varphi}\right|}d\varphi \tag{8}$$

$$= \frac{1}{2\pi}\int_{-\pi}^{\pi}\arccos\frac{\text{Re}\left(\frac{\varepsilon}{|\hat{Y}|}\cdot(\cos(\varphi)+i\cdot\sin(\varphi))+1\right)}{\left|\frac{\varepsilon}{|\hat{Y}|}+\cos(\varphi)+i\cdot\sin(\varphi)\right|}d\varphi \tag{9}$$

$$= \frac{1}{2\pi}\int_{-\pi}^{\pi}\arccos\frac{\frac{\varepsilon\cdot\cos(\varphi)}{|\hat{Y}|}+1}{\sqrt{\left(\frac{\varepsilon}{|\hat{Y}|}+\cos(\varphi)\right)^2+\sin^2(\varphi)}}d\varphi \tag{10}$$

$$= \frac{1}{2\pi}\int_{-\pi}^{\pi}\arccos\frac{\frac{\varepsilon\cdot\cos(\varphi)}{|\hat{Y}|}+1}{\sqrt{\left(\frac{\varepsilon}{|\hat{Y}|}\right)^2+\frac{2\varepsilon\cdot\cos(\varphi)}{|\hat{Y}|}+1}}d\varphi \tag{11}$$

$$= \frac{1}{2\pi}\int_{-\pi}^{\pi}\arccos\left[\left(\frac{\varepsilon\cdot\cos(\varphi)}{|\hat{Y}|}+1\right)\cdot\left(\frac{1}{\sqrt{\left(\frac{\varepsilon}{|\hat{Y}|}\right)^2+\frac{2\varepsilon\cdot\cos(\varphi)}{|\hat{Y}|}+1}}\right)\right]d\varphi \tag{12}$$

Assume that we are in high error regime, i.e. $\frac{\varepsilon}{|\hat{Y}|}\gg 1$:

$$\mathbb{E}_Y\left(\mathcal{L}^{(\text{phase})}(Y,\hat{Y})\right) \approx \frac{1}{2\pi}\int_{-\pi}^{\pi}\arccos\left[\frac{\frac{\varepsilon}{|\hat{Y}|}\cos(\varphi)}{\sqrt{\left(\frac{\varepsilon}{|\hat{Y}|}\right)^2+\frac{2\varepsilon\cdot\cos(\varphi)}{|\hat{Y}|}+1}}\right]d\varphi \tag{13}$$

Since in the high-error regime where $\frac{\varepsilon}{|\hat{Y}|}\gg 1$ the constant term 1 in the numerator can be disregarded as negligible. Then $\mathbb{E}_Y\left(\mathcal{L}^{(\text{phase})}(Y,\hat{Y})\right)$ can be written as:

$$\frac{1}{2\pi}\int_{-\pi}^{\pi}\arccos\left[\frac{\cos(\varphi)}{\sqrt{1+\frac{2|\hat{Y}|\cdot\cos(\varphi)}{\varepsilon}+\left(\frac{|\hat{Y}|}{\varepsilon}\right)^2}}\right]d\varphi \tag{14}$$

$$\approx \frac{1}{2\pi}\int_{-\pi}^{\pi}\arccos\left[\frac{\cos(\varphi)}{\sqrt{1+\frac{2|\hat{Y}|\cdot\cos(\varphi)}{\varepsilon}}}\right]d\varphi \tag{15}$$

$$\approx \frac{1}{2\pi}\int_{-\pi}^{\pi}\arccos\left[\cos(\varphi)\left(1-\frac{|\hat{Y}|\cdot\cos(\varphi)}{\varepsilon}\right)\right]d\varphi \tag{16}$$

Since, in high error regime $\left(\frac{|\hat{Y}|}{\varepsilon}\right)^2 \ll 1$ and the Taylor series expansion employed is $\frac{1}{\sqrt{1+x}} \approx 1 - \frac{x}{2}$ where $x = \frac{2|\hat{Y}| \cdot \cos(\varphi)}{\varepsilon}$. Thus, $\mathbb{E}_Y\left(\mathcal{L}^{(\text{phase})}(Y, \hat{Y})\right)$ can be expressed as:

$$\frac{1}{2\pi} \int_{-\pi}^{\pi} \arccos\left[\cos(\varphi) - \frac{|\hat{Y}|}{\varepsilon} \cdot \cos^2(\varphi)\right] d\varphi \tag{17}$$

$$= \frac{1}{2\pi} \int_{-\pi}^{\pi} \arccos\left[\cos(\varphi) - \frac{|\hat{Y}|}{\varepsilon} \cdot \left(\frac{\cos(2\varphi) + 1}{2}\right)\right] d\varphi \tag{18}$$

$$= \frac{1}{2\pi} \int_{-\pi}^{\pi} \arccos\left[\cos(\varphi) - \frac{|\hat{Y}|}{\varepsilon} \cdot \frac{\cos(2\varphi)}{2} - \frac{|\hat{Y}|}{2\varepsilon}\right] d\varphi \tag{19}$$

$$\approx \frac{1}{2\pi} \int_{-\pi}^{\pi} \arccos\left[\cos(\varphi) - \frac{|\hat{Y}|}{\varepsilon} \cdot \frac{\cos(2\varphi)}{2}\right] d\varphi \tag{20}$$

where $\frac{|\hat{Y}|}{\varepsilon}$ can be neglected as $\frac{|\hat{Y}|}{\varepsilon} \ll 1$. The Taylor Series expansion $\arccos(x) \approx \frac{\pi}{2} - x$ is used, where $x = \cos(\varphi) - \frac{|\hat{Y}|}{\varepsilon} \cdot \frac{\cos(2\varphi)}{2}$. Therefore, $\mathbb{E}_Y\left(\mathcal{L}^{(\text{phase})}(Y, \hat{Y})\right)$ is equal to:

$$\frac{1}{2\pi} \int_{-\pi}^{\pi} \frac{\pi}{2} - \cos(\varphi) + \frac{|\hat{Y}|}{\varepsilon} \cdot \frac{\cos(2\varphi)}{2} d\varphi \tag{21}$$

$$= \frac{1}{2\pi} \int_{-\pi}^{\pi} \left(\frac{\pi}{2} - \cos(\varphi) + \frac{|\hat{Y}|}{\varepsilon} \cdot \frac{\cos(2\varphi)}{2}\right) d\varphi \tag{22}$$

$$= \frac{1}{2\pi} \int_{-\pi}^{\pi} \frac{\pi}{2} d\varphi + \frac{1}{2\pi} \int_{-\pi}^{\pi} \cos(\varphi) d\varphi + \frac{1}{2\pi} \int_{-\pi}^{\pi} \frac{|\hat{Y}|}{\varepsilon} \cdot \frac{\cos(2\varphi)}{2} d\varphi \tag{23}$$

$$= \frac{\pi}{2} + 0 + 0 = \frac{\pi}{2} \tag{24}$$

Overall, the phase error is expressed as:

$$\mathbb{E}_Y\left(\mathcal{L}^{(\text{phase})}(Y, \hat{Y})\right) \approx \frac{\pi}{2}. \tag{25}$$

$\square$

### E.2 AMPLITUDE ERROR:

We can then start from the definition of the amplitude error from lemma 1 in (Richard et al., 2021) and solve the integral:

$$\mathbb{E}_Y\left(\mathcal{L}^{(\text{amp})}(Y,\hat{Y})\right) = \frac{|\hat{Y}|}{2\pi}\int_{-\pi}^{\pi}\left||\frac{\varepsilon}{|\hat{Y}|}+e^{i\varphi}|-1\right|d\varphi \tag{26}$$

$$= \frac{|\hat{Y}|}{2\pi}\int_{-\pi}^{\pi}\left|\sqrt{(\frac{\varepsilon}{|\hat{Y}|}+\cos\varphi)^2+\sin^2\varphi}-1\right|d\varphi \tag{27}$$

$$= \frac{|\hat{Y}|}{2\pi}\int_{-\pi}^{\pi}\left|\sqrt{\left(\frac{\varepsilon}{|\hat{Y}|}\right)^2+\frac{2\varepsilon\cos\varphi}{|\hat{Y}|}+\cos^2\varphi+\sin^2\varphi}-1\right|d\varphi \tag{28}$$

$$= \frac{|\hat{Y}|}{2\pi}\int_{-\pi}^{\pi}\left|\sqrt{\left(\frac{\varepsilon}{|\hat{Y}|}\right)^2+\frac{2\varepsilon\cos\varphi}{|\hat{Y}|}+1}-1\right|d\varphi \tag{29}$$

$$= \frac{|\hat{Y}|}{2\pi}\int_{-\pi}^{\pi}\left|\frac{\varepsilon}{|\hat{Y}|}\sqrt{1+\frac{2|\hat{Y}|\cos\varphi}{\varepsilon}+\left(\frac{|\hat{Y}|}{\varepsilon}\right)^2}-1\right|d\varphi \tag{30}$$

$$\overset{(*)}{\approx} \frac{|\hat{Y}|}{2\pi}\int_{-\pi}^{\pi}\left|\frac{\varepsilon}{|\hat{Y}|}\sqrt{1+\frac{2|\hat{Y}|\cos\varphi}{\varepsilon}}-1\right|d\varphi \tag{31}$$

$$\overset{(**)}{\approx} \frac{|\hat{Y}|}{2\pi}\int_{-\pi}^{\pi}\left|\frac{\varepsilon}{|\hat{Y}|}\left(1+\frac{1}{2}\cdot\frac{2|\hat{Y}|\cos\varphi}{\varepsilon}\right)-1\right|d\varphi \tag{32}$$

$$= \frac{|\hat{Y}|}{2\pi}\int_{-\pi}^{\pi}\left|\frac{\varepsilon}{|\hat{Y}|}+\cos\varphi-1\right|d\varphi \tag{33}$$

$$\overset{(***)}{\approx} \frac{|\hat{Y}|}{2\pi}\int_{-\pi}^{\pi}\left(\frac{\varepsilon}{|\hat{Y}|}+\cos\varphi\right)d\varphi \tag{34}$$

$$= \frac{|\hat{Y}|}{2\pi}\cdot\frac{\varepsilon}{|\hat{Y}|}\cdot 2\pi+\frac{|\hat{Y}|}{2\pi}\int_{-\pi}^{\pi}\cos\varphi d\varphi \tag{35}$$

$$= \varepsilon+0=\varepsilon \tag{36}$$

In the above derivation, the following approximations were employed, under the assumption that $\frac{\varepsilon}{|\hat{Y}|}\gg 1$:

1. $(*)$ Removing the term $\left(\frac{|\hat{y}|}{\varepsilon}\right)^2$ since by the assumption it is negligible.

2. $(**)$ Using the Taylor Series expansion: $\sqrt{1+x}\approx 1+\frac{x}{2}$ where $x=\frac{2|\hat{Y}|\cdot\cos(\varphi)}{\varepsilon}$

3. $(***)$ Removing the term $1$ and the $|\cdot|$ fucntion since the overall integrand is dominate by the term $\frac{\varepsilon}{|\hat{Y}|}$.

Overall, the amplitude error is expressed as - $\mathbb{E}_Y\left(\mathcal{L}^{(\text{amp})}(Y,\hat{Y})\right)\approx\varepsilon$. $\square$

## F  LOW-ERROR REGIME (LEMMA 2)

**Lemma 2.** *Let $\hat{Y} \in \mathbb{C}$, and let there be a ball of complex numbers with distance $\varepsilon$ from $\hat{Y}$ such that $Y \in \mathbb{B}_\varepsilon = \{Y \in \mathbb{C} : |Y - \hat{Y}| = \varepsilon\}$. Assuming a low error regime where $\frac{\varepsilon}{|\hat{Y}|} \ll 1$, then the expected amplitude and phase errors are:*

$$\mathbb{E}_Y\left(\mathcal{L}^{(phase)}(Y, \hat{Y})\right) \approx \left(\frac{\pi}{2} - 1\right) + \frac{\varepsilon^2}{2|\hat{Y}|^2}, \tag{37}$$

$$\mathbb{E}_Y\left(\mathcal{L}^{(amp)}(Y, \hat{Y})\right) \approx \begin{cases} \varepsilon - \dfrac{\pi^2 |\hat{Y}|\varepsilon}{3(2|\hat{Y}|+\varepsilon)} & , \dfrac{\varepsilon}{|\hat{Y}|} \geq \dfrac{\pi^2}{2} - 1 \\[4mm] \dfrac{\pi^2 |\hat{Y}|\varepsilon}{3(2|\hat{Y}|+\varepsilon)} - \varepsilon + \dfrac{4\varepsilon\sqrt{\frac{2|\hat{Y}|+\varepsilon}{|\hat{Y}|\varepsilon}}}{3\pi} & , \dfrac{\varepsilon}{|\hat{Y}|} \leq \dfrac{\pi^2}{2} - 1. \end{cases} \tag{38}$$

*Proof.* **Angular phase error:** We can then start from the definition of the phase error from lemma 1 in (Richard et al., 2021) and solve the integral:

$$\mathbb{E}_Y\left(\mathcal{L}^{(phase)}(Y, \hat{Y})\right) = \frac{1}{2\pi} \int_{-\pi}^{\pi} \arccos \frac{\text{Re}\left(\frac{\varepsilon}{|\hat{Y}|} \cdot e^{i\varphi} + 1\right)}{\left|\frac{\varepsilon}{|\hat{Y}|} + e^{i\varphi}\right|} d\varphi \tag{39}$$

$$= \frac{1}{2\pi} \int_{-\pi}^{\pi} \arccos \frac{\text{Re}\left(\frac{\varepsilon}{|\hat{Y}|} \cdot (\cos(\varphi) + i \cdot \sin(\varphi)) + 1\right)}{\left|\frac{\varepsilon}{|\hat{Y}|} + \cos(\varphi) + i \cdot \sin(\varphi)\right|} d\varphi \tag{40}$$

$$= \frac{1}{2\pi} \int_{-\pi}^{\pi} \arccos \frac{\frac{\varepsilon \cdot \cos(\varphi)}{|\hat{Y}|} + 1}{\sqrt{\left(\frac{\varepsilon}{|\hat{Y}|} + \cos(\varphi)\right)^2 + \sin^2(\varphi)}} d\varphi \tag{41}$$

Since $\cos^2(\varphi) + \sin^2(\varphi) = 1$, the phase error $\mathbb{E}_Y\left(\mathcal{L}^{(phase)}(Y, \hat{Y})\right)$ can be expressed as:

$$= \frac{1}{2\pi} \int_{-\pi}^{\pi} \arccos \frac{\frac{\varepsilon \cdot \cos(\varphi)}{|\hat{Y}|} + 1}{\sqrt{\left(\frac{\varepsilon}{|\hat{Y}|}\right)^2 + \frac{2\varepsilon \cdot \cos(\varphi)}{|\hat{Y}|} + 1}} d\varphi \tag{42}$$

$$= \frac{1}{2\pi} \int_{-\pi}^{\pi} \arccos \left[\left(\frac{\varepsilon \cdot \cos(\varphi)}{|\hat{Y}|} + 1\right) \cdot \left(\frac{1}{\sqrt{\left(\frac{\varepsilon}{|\hat{Y}|}\right)^2 + \frac{2\varepsilon \cdot \cos(\varphi)}{|\hat{Y}|} + 1}}\right)\right] d\varphi \tag{43}$$

$$\approx \frac{1}{2\pi} \int_{-\pi}^{\pi} \arccos \left(\frac{\varepsilon \cdot \cos(\varphi)}{|\hat{Y}|} + 1\right) \cdot \left(1 - \frac{1}{2}\left(\frac{\varepsilon}{|\hat{Y}|}\right)^2 - \frac{\varepsilon \cdot \cos(\varphi)}{|\hat{Y}|}\right) d\varphi \tag{44}$$

$$\approx \frac{1}{2\pi} \int_{-\pi}^{\pi} \arccos \left(1 + \frac{\varepsilon \cdot \cos(\varphi)}{|\hat{Y}|}\right) \cdot \left(1 - \frac{\varepsilon \cdot \cos(\varphi)}{|\hat{Y}|}\right) d\varphi \tag{45}$$

Utilizing Taylor expansion $\frac{1}{\sqrt{1+x}} \approx 1 - \frac{x}{2}$ when $x = \left(\frac{\varepsilon}{|\hat{Y}|}\right)^2 + \frac{2\varepsilon \cdot \cos(\varphi)}{|\hat{Y}|}$ and removing the term $\frac{1}{2}\left(\frac{\varepsilon}{|\hat{y}|}\right)^2$ since by our assumption it is negligible. Therefore, the phase error $\mathbb{E}_Y\left(\mathcal{L}^{(phase)}(Y, \hat{Y})\right)$ can be written as:

$$= \frac{1}{2\pi} \int_{-\pi}^{\pi} \arccos \left(1 - \frac{\varepsilon^2 \cdot \cos^2(\varphi)}{|\hat{Y}|^2}\right) d\varphi \tag{46}$$

$$\overset{(***)}{\approx} \frac{1}{2\pi} \int_{-\pi}^{\pi} \left(\frac{\pi}{2} - 1 + \frac{\varepsilon^2 \cdot \cos^2(\varphi)}{|\hat{Y}|^2}\right) d\varphi \tag{47}$$

Since $\arccos(x) \approx \frac{\pi}{2} - x$ where $x = 1 - \frac{\varepsilon^2 \cdot \cos^2 \varphi}{|\hat{Y}|^2}$. Then, the phase error $\mathbb{E}_Y\left(\mathcal{L}^{(\text{phase})}(Y, \hat{Y})\right)$ can be expressed as:

$$= \left(\frac{\pi}{2} - 1\right) + \frac{1}{2\pi} \int_{-\pi}^{\pi} \frac{\varepsilon^2 \cdot \cos^2(\varphi)}{|\hat{Y}|^2} d\varphi \tag{48}$$

$$= \left(\frac{\pi}{2} - 1\right) + \frac{\varepsilon^2}{2\pi|\hat{Y}|^2} \int_{-\pi}^{\pi} \cos^2(\varphi) d\varphi \tag{49}$$

$$= \left(\frac{\pi}{2} - 1\right) + \frac{\varepsilon^2}{2\pi|\hat{Y}|^2} \int_{-\pi}^{\pi} \cos^2(\varphi) d\varphi \tag{50}$$

$$= \left(\frac{\pi}{2} - 1\right) + \frac{\varepsilon^2}{2\pi|\hat{Y}|^2} \left[\frac{\varphi}{2} + \frac{sin(2\varphi)}{4}\right]\Big|_{-\pi}^{\pi} \tag{51}$$

$$= \left(\frac{\pi}{2} - 1\right) + \frac{\varepsilon^2}{2|\hat{Y}|^2} \tag{52}$$

**Amplitude error:** We can then start from the definition of the amplitude error from lemma 1 in (Richard et al., 2021) and solve the integral:

$$
\mathbb{E}_Y\left(\mathcal{L}^{(\text{amp})}(Y,\hat{Y})\right) = \frac{|\hat{Y}|}{2\pi}\int_{-\pi}^{\pi}\left|\,|\frac{\varepsilon}{|\hat{Y}|} + e^{i\varphi}| - 1\right|d\varphi
\tag{53}
$$

$$
= \frac{|\hat{Y}|}{2\pi}\int_{-\pi}^{\pi}\left|\sqrt{(\frac{\varepsilon}{|\hat{Y}|} + \cos\varphi)^2 + \sin^2\varphi} - 1\right|d\varphi
\tag{54}
$$

$$
= \frac{|\hat{Y}|}{2\pi}\int_{-\pi}^{\pi}\left|\sqrt{\left(\frac{\varepsilon}{|\hat{Y}|}\right)^2 + \frac{2\varepsilon\cos\varphi}{|\hat{Y}|} + \cos^2\varphi + \sin^2\varphi} - 1\right|d\varphi
\tag{55}
$$

$$
= \frac{|\hat{Y}|}{2\pi}\int_{-\pi}^{\pi}\left|\sqrt{\left(\frac{\varepsilon}{|\hat{Y}|}\right)^2 + \frac{2\varepsilon\cos\varphi}{|\hat{Y}|} + 1} - 1\right|d\varphi
\tag{56}
$$

$$
= \frac{|\hat{Y}|}{2\pi}\int_{-\pi}^{\pi}\left|\sqrt{\left(\frac{\varepsilon}{|\hat{Y}|}\right)^2 + 1 + \frac{2\varepsilon\cos\varphi}{|\hat{Y}|}} - 1\right|d\varphi
\tag{57}
$$

$$
\overset{*}{\approx} \frac{|\hat{Y}|}{2\pi}\int_{-\pi}^{\pi}\left|\sqrt{\left(\frac{\varepsilon}{|\hat{Y}|}\right)^2 + 1 + \frac{2\varepsilon}{|\hat{Y}|}} - \frac{\frac{2\varepsilon\varphi^2}{|\hat{Y}|}}{4\sqrt{\left(\frac{\varepsilon}{|\hat{Y}|}\right)^2 + 1 + \frac{2\varepsilon}{|\hat{Y}|}}} - 1\right|d\varphi
\tag{58}
$$

$$
= \frac{|\hat{Y}|}{2\pi}\int_{-\pi}^{\pi}\left|1 + \frac{\varepsilon}{|\hat{Y}|} - \frac{\frac{2\varepsilon\varphi^2}{|\hat{Y}|}}{4\left(1 + \frac{\varepsilon}{|\hat{Y}|}\right)} - 1\right|d\varphi
\tag{59}
$$

$$
= \frac{|\hat{Y}|}{2\pi}\int_{-\pi}^{\pi}\left|\frac{\varepsilon}{|\hat{Y}|} - \frac{\frac{2\varepsilon\varphi^2}{|\hat{Y}|}}{4\left(1 + \frac{\varepsilon}{|\hat{Y}|}\right)}\right|d\varphi
\tag{60}
$$

$$
= \frac{|\hat{Y}|}{2\pi}\cdot\frac{\varepsilon}{|\hat{Y}|}\int_{-\pi}^{\pi}\left|1 - \frac{2\varphi^2}{4\left(1 + \frac{\varepsilon}{|\hat{Y}|}\right)}\right|d\varphi
\tag{61}
$$

$$
= \frac{\varepsilon}{2\pi}\int_{-\pi}^{\pi}\left|1 - \frac{\varphi^2}{2\left(1 + \frac{\varepsilon}{|\hat{Y}|}\right)}\right|d\varphi
\tag{62}
$$

$$
= \frac{\varepsilon}{2\pi}\frac{1}{2\left(1 + \frac{\varepsilon}{|\hat{Y}|}\right)}\int_{-\pi}^{\pi}\left|2\left(1 + \frac{\varepsilon}{|\hat{Y}|}\right) - \varphi^2\right|d\varphi
\tag{63}
$$

We can then write

$$
a = 2\left(1 + \frac{\varepsilon}{2|\hat{Y}|}\right),\ \frac{\varepsilon}{2a\pi}\int_{-\pi}^{\pi}\left|a - \varphi^2\right|d\varphi
\tag{64}
$$

And thus re-write the amplitude error as:

$$
\mathbb{E}_Y\left(\mathcal{L}^{(\text{amp})}(Y,\hat{Y})\right) = \frac{\varepsilon}{2a\pi}\int_{-\pi}^{\pi}\left|a - \varphi^2\right|d\varphi
\tag{65}
$$

The final error function will be a split function between $a > \pi^2$ and $a \le \pi^2$. For $a > \pi^2$ we write:

$$\mathbb{E}_Y\left(\mathcal{L}^{(\text{amp})}(Y, \hat{Y})\right) = \frac{\varepsilon}{2a\pi} \int_{-\pi}^{\pi} \left|a - \varphi^2\right| d\varphi \tag{66}$$

$$= \frac{\varepsilon}{2a\pi}\left(a\varepsilon - \frac{\varepsilon^3}{3}\right)\Big|_{-\pi}^{\pi} = \frac{\varepsilon}{2a\pi}\left(2\pi a - \frac{2\pi^3}{3}\right) \tag{67}$$

$$= \varepsilon\left(1 - \frac{\pi^2}{3a}\right) = \varepsilon - \frac{\varepsilon\pi^2}{3a} \tag{68}$$

$$= \varepsilon - \frac{\varepsilon\pi^2}{6\left(1 + \frac{\varepsilon}{2|\hat{Y}|}\right)} = \varepsilon - \frac{\pi^2}{6\left(\frac{1}{\varepsilon} + \frac{1}{2|\hat{Y}|}\right)} \tag{69}$$

$$= \varepsilon - \frac{\pi^2}{6\left(\frac{2|\hat{Y}|+\varepsilon}{2|\hat{Y}|\varepsilon}\right)} = \varepsilon - \frac{\pi^2|\hat{Y}|\varepsilon}{3(2|\hat{Y}| + \varepsilon)} \tag{70}$$

For $a \le \pi^2$ we can write:

$$\mathbb{E}_Y\left(\mathcal{L}^{(\text{amp})}(Y, \hat{Y})\right) = \frac{\varepsilon}{2a\pi} \int_{-\pi}^{\pi} \left|a - \varphi^2\right| d\varphi \tag{71}$$

$$= \frac{\varepsilon}{2a\pi}\left[\int_{-\pi}^{-\sqrt{a}} \left(\varphi^2 - a\right) d\varphi + \int_{-\sqrt{a}}^{\sqrt{a}} \left(a - \varphi^2\right) d\varphi + \int_{\sqrt{a}}^{\pi} \left(\varphi^2 - a\right) d\varphi\right] \tag{72}$$

$$= \frac{\varepsilon}{2a\pi}\left[\int_{-\sqrt{a}}^{\sqrt{a}} \left(a - \varphi^2\right) d\varphi + 2\int_{\sqrt{a}}^{\pi} \left(\varphi^2 - a\right) d\varphi\right] \tag{73}$$

$$= \frac{\varepsilon}{2a\pi}\left[\left(a\varphi - \frac{\varphi^3}{3}\right)\Big|_{-\sqrt{a}}^{\sqrt{a}} + 2\left(\frac{\varphi^3}{3} - a\varphi\right)\Big|_{\sqrt{a}}^{\pi}\right] \tag{74}$$

$$= \frac{\varepsilon}{2a\pi}\left[2\left(a^{3/2} - \frac{a^{3/2}}{3}\right) + 2\left(\frac{\pi^3 - a^{3/2}}{3} - a(\pi - \sqrt{a})\right)\right] \tag{75}$$

$$= \frac{\varepsilon}{2a\pi}\left[2\left(a^{3/2} - \frac{a^{3/2}}{3}\right) + 2\left(\frac{\pi^3 - a^{3/2}}{3} - a(\pi - \sqrt{a})\right)\right] \tag{76}$$

$$= \frac{\varepsilon}{2a\pi}\left[\frac{2\pi^3}{3} - 2\pi a + \frac{8a^{3/2}}{3}\right] \tag{77}$$

$$= \varepsilon\left[\frac{\pi^2}{3a} - 1 + \frac{4a^{1/2}}{3\pi}\right] \tag{78}$$

$$= \frac{\varepsilon\pi^2}{3a} - \varepsilon + \frac{4a^{1/2}\varepsilon}{3\pi} \tag{79}$$

$$= \frac{\varepsilon\pi^2}{6\left(1 + \frac{\varepsilon}{2|\hat{Y}|}\right)} - \varepsilon + \frac{4\varepsilon\sqrt{2\left(1 + \frac{\varepsilon}{2|\hat{Y}|}\right)}}{3\pi} \tag{80}$$

$$= \frac{\pi^2}{6\left(\frac{2|\hat{Y}|+\varepsilon}{2|\hat{Y}|\varepsilon}\right)} - \varepsilon + \frac{4\varepsilon\sqrt{2\left(\frac{2|\hat{Y}|+\varepsilon}{2|\hat{Y}|\varepsilon}\right)}}{3\pi} \tag{81}$$

$$= \frac{\pi^2|\hat{Y}|\varepsilon}{3(2|\hat{Y}| + \varepsilon)} - \varepsilon + \frac{4\varepsilon\sqrt{\frac{2|\hat{Y}|+\varepsilon}{|\hat{Y}|\varepsilon}}}{3\pi} \tag{82}$$

Finally, we can merge the results from both the phase and amplitude errors to get

$$\mathbb{E}_Y\left(\mathcal{L}^{(\text{phase})}(Y,\hat{Y})\right) \approx \left(\frac{\pi}{2}-1\right) + \frac{\varepsilon^2}{2|\hat{Y}|^2} \tag{83}$$

$$\mathbb{E}_Y\left(\mathcal{L}^{(\text{amp})}(Y,\hat{Y})\right) \approx \begin{cases} \varepsilon - \frac{\pi^2|\hat{Y}|\varepsilon}{3(2|\hat{Y}|+\varepsilon)} & , 2\left(1+\frac{\varepsilon}{|\hat{Y}|}\right) > \pi^2 \\ \frac{\pi^2|\hat{Y}|\varepsilon}{3(2|\hat{Y}|+\varepsilon)} - \varepsilon + \frac{4\varepsilon\sqrt{\frac{2|\hat{Y}|+\varepsilon}{|\hat{Y}|\varepsilon}}}{3\pi} & , 2\left(1+\frac{\varepsilon}{|\hat{Y}|}\right) \le \pi^2 \end{cases} \tag{84}$$

$\square$

