# OpenReview forum: "Towards Universal Mono-to-Binaural Speech Synthesis"
_ICLR.cc/2025/Conference — ICLR 2025 Conference Withdrawn Submission_

### Official Review · Reviewer_jX7X · 2024-10-31

**Soundness:** 4
**Presentation:** 3
**Contribution:** 3
**Rating:** 5
**Confidence:** 5

**Summary:**

The paper addresses the challenge of converting monaural speech audio into binaural audio for arbitrary environments, which is crucial for applications in telepresence and extended reality. The authors identify that current neural mono-to-binaural methods overfit to non-spatial acoustic properties, as revealed by a new benchmark dataset (TUT Mono-to-Binaural). To overcome this, they propose BinauralZero, a baseline model for universal mono-to-binaural synthesis that does not require any binaural data for training. BinauralZero leverages simple parameter-free transforms and an off-the-shelf denoising vocoder to enhance the initial binauralization. The authors demonstrate that BinauralZero can match or outperform state-of-the-art neural mono-to-binaural renderers in subjective quality despite its simplicity.

**Strengths:**

1. Novel Benchmark Dataset: The introduction of the TUT Mono-to-Binaural dataset is a significant contribution, providing a more diverse and challenging benchmark for evaluating mono-to-binaural synthesis methods.

2. Universality and Generalization: BinauralZero's ability to perform well across different acoustic environments without specific training data is a substantial advantage, showcasing its potential for universal application.

3. Simplicity and Effectiveness: The proposed method's reliance on simple, parameter-free transforms combined with an off-the-shelf denoising model results in a system that is both easy to implement and effective, which is a practical strength for real-world applications.

**Weaknesses:**

1. There seems to be some over-claims. The methods proposed in paper is more like a method that firstly using DSP method (warping and scaling) to change the mono audio to binaural audio. And then using a vocoder to refine the audio quality. The claim in L262 is hard to understand. Why the last step in diffusion based vocoder can model the RIR and HRTF? For claim in L252, why the positional information can be conditioned implicitly by providing an almost-complete waveform? The positional is a relative concept in binaural audio. So at least the model need take the binaural audio as input and then it can implicitly condition on position. However, in the BinauralZero, the vocoder only takes mono audio. So it is more like a waveform refiner.

2. I agree with the motivation in the paper that the network based supervised model tends to overfit when the training dataset is small. However, the proposed method should be compared with more DSP based models (e.g., some models with general RIR and HRTF) to show the effectiveness of proposed method on generalization.

**Questions:**

1. In vocoder, will the output from DSP directly be fed into the vocoder or firstly adding noise (at the low level corresponding to the step) and then be fed into the vocoder?

2. In table 4, why removing AS causes significantly drop on N-MOS, which is quite strange.

---

### Official Review · Reviewer_mB2J · 2024-10-31

**Soundness:** 1
**Presentation:** 2
**Contribution:** 1
**Rating:** 3
**Confidence:** 4

**Summary:**

- The paper releases a new benchmark dataset for the Mono-to-Binaural synthesis task, which is synthetic, binaural, and ambisonic rendered speeches.
- It proposes a framework named BinauralZero, which applies delay and attenuation corresponding to the shortest path between the source and receivers, converts it into a Mel spectrogram, and then applies a diffusion-based neural network vocoder to convert it back into the waveform.
- It reports subjective evaluation results to show that BinauralZero can outperform the baselines without training with binaural datasets, building on top of pre-trained off-the-shelf neural vocoders.

**Strengths:**

The authors highlight various aspects between the objective and the subjective scores prevalently used in the binaural synthesis literature.

**Weaknesses:**

Overall, the authors tend to trust subjective metrics (that they've done) over objective metrics and draw conclusions based on them, but the more I think about it, the more questions I have about the subjective evaluation process.
Also for all three of the main contributions: not enough logical connections were explained or consensus was reached to support the claims.

1. There is no analysis of the dataset they propose and the authors' description is misleading.
    - Throughout the paper, the TUT dataset is described as anechoic (L138, L221, L449), and I believed that for a while, but after actually listening to the GTs in the supplementary material, I realized that they are very echoic. Please clarify this.

2. In fact, whether the TUT dataset is reverberant or anechoic, both cases raise questions about the novelty of this work.
    1. If the TUT dataset is anechoic (and the GT samples in the supplementary material are incorrectly attached), it seems self-evident that BinauralZero will perform better on anechoic datasets.
        - The baselines were trained on binaural 'room' recordings, so they will always contain room reverb. BinauralZero, on the other hand, will follow the recording environments of the training data of its vocoder (WaveFit), and although it is not stated in the WaveFit paper, given that the test data is LibriTTS, it seems likely that there is little room characterization in the training data. (In fact, if one listens to BinauralZero's output on the TUT data, the reverb is very weak.) Based on this fact alone, it is not surprising that BinauralZero outperforms the baselines on the TUT data, given that TUT is an 'anechoic binaural rendering'. How can this result be related to the claim that “the existing neural models seem to highly overfit to non-spatial acoustic features”? Please provide any specific examples of ‘non-spatial acoustic features’ that the existing neural models seem to highly overfit.

    2. Conversely, if the TUT dataset is reverberant and the GT samples in the supplementary material are all properly attached, how could the subjective assessment result in a high MUSHRA score despite BinauralZero being anechoic? The fact that the model rated most similar to the (reverberant) TUT GT is the (anechoic) BinauralZero does not resonate with my consensus. Other than this:
        1. One can clearly hear out-of-phase artifacts from most of BinauralZero’s output for the Binaural Speech dataset (especially for the sibilant sounds). On the other hand, those artifacts are hard to find from the outputs for the TUT dataset. Please elaborate on this discrepancy.
        2. NFS outputs for the TUT dataset are slightly detuned. As far as I know, NFS only applies multichannel linear phased filters (so it cannot change a pitch by its design.) Please expand on this.
        3. All of the baseline models were trained on a 48 kHz sampling rate, WaveFit is trained on a 24 kHz sampling rate, and the TUT dataset seems to have a 44.1 kHz sampling rate. Nothing is mentioned about the conversions between these differences.
        4. All of the baseline models require orientation to be input. How were the orientations of the sources in the TUT dataset synthesized? What, if any, differences in the distribution of orientation and location coordinates are there from the Binaural Speech dataset, and what are the implications?
        5. Tests primarily using MUSHRA will also include results for hidden anchors. What are the anchors in this paper? How were they scored?

3. The ablations seem to deserve better experiment setup, as so many questions arise:
    1. If one is to insist that the four “automated metrics” mentioned in the paper decorrelate to perceptual metrics, why not report other metrics (e.g., SDR, SAQAM [1], NORD [2], AODG [3])? NORD is even open-sourced.
    2. Provide the configurations for the mel-spectrogram conversion (window/hop size, number of FFT points, number of Mels, …). Then, please clarify how BinauralZero can model the accurate delay within the hop size (12.5 ms).
        - A typical maximum ITD is considered 0.66 ms (when a sound source is positioned at 90° azimuth to one ear), but let's say the ITD for the GTW's output was 1 ms for the sake of brevity (this amounts to approximately for the maximal difference in time of arrival for the subject with 40 cm distance between ears). In the BinauralZero framework, the waveform with 1ms ITD will be converted into a Mel spectrogram to be generated as a 'natural-sounding waveform' via vocoder. How will this 1ms ITD be preserved throughout this process?

[1] Manocha, P., Kumar, A., Xu, B., Menon, A., Gebru, I. D., Ithapu, V. K., & Calamia, P. (2022). SAQAM: Spatial audio quality assessment metric. *arXiv preprint arXiv:2206.12297*.

[2] Manocha, P., Gebru, I. D., Kumar, A., Markovic, D., & Richard, A. (2023, June). Nord: Non-matching reference based relative depth estimation from binaural speech. In *ICASSP 2023-2023 IEEE International Conference on Acoustics, Speech and Signal Processing (ICASSP)* (pp. 1-5). IEEE.

[3] Schäfer, M., Bahram, M., & Vary, P. (2013, May). An extension of the PEAQ measure by a binaural hearing model. In *2013 IEEE International Conference on Acoustics, Speech and Signal Processing* (pp. 8164-8168). IEEE.

**Questions:**

1. Can you expand more audio domain-specific details about the synthesis process of TUT?
    1. What kind of head-related transfer functions (HRTFs) did you use for synthesizing TUT?
    2. How does the transfer function change in terms of distance/azimuth/elevation changes?
    3. Does it incorporate any linear/nonlinear systems?
2. Regarding GTW, there are many other interpolations to feed the delay of non-integer values, one could apply a linear phase filter like NFS. Is there a reason why you chose linear interpolation?
3. Regarding AS, what do you mean by “we are the first to explicitly apply to neural mono-to-binaural synthesis”? How is it different from NFS’s amplitude scaling?
4. Can you elaborate justifications on the choice of
    1. GTW+AS
        - Please compare using “Resonance Audio package” (the DSP baseline, instead of GTW+AS) along with WaveFit?
    2. Vocoder
        - Please compare using Griffin-Lim, BigVGAN [4], etc.
5. Please compare the computational complexities along with the scores.

[4] Lee, S. G., Ping, W., Ginsburg, B., Catanzaro, B., & Yoon, S. BigVGAN: A Universal Neural Vocoder with Large-Scale Training. In The Eleventh International Conference on Learning Representations.

---

### Official Review · Reviewer_QDf5 · 2024-11-04

**Soundness:** 2
**Presentation:** 3
**Contribution:** 3
**Rating:** 5
**Confidence:** 4

**Summary:**

- A new parameter-free mono-to-binaural method, which consists of geometric time warping, amplitude scaling, and denoising, is proposed.
- A new benchmark TUT Mono-to-Binaural is proposed to evaluate the model performance.
- Both subjective and objective evaluations are conducted. The proposed method performs well for subjective metrics both on Binaural Speech and TUT Mono-to-Binaural. For objective metrics, it performs well on TUT Mono-to-Binaural but not on Bonural Speech.
- Further experiments show that the objective metrics may not reflect the perceptual quality of the generated binaural speech, or even be misleading.
- Future work includes modeling phase and integrating room or head information.

**Strengths:**

- The proposed method is parameter-free and easy to follow, and it can be a plug-in module for other methods. Although some components are off-the-shelf or have been applied in previous work, the integration of different components is reasonable. The ablation study demonstrates the effectiveness of each module.
- The proposed TUT Mono-to-Binaural makes an addition to the community and it serves as an extra evaluation benchmark.
- The theoretical induction for both the phase and amplitude error are presented to explain the experimental results. Figure 3 shows that the conditions of a high error regime are satisfied.

**Weaknesses:**

- The comparison for baseline methods on TUT Mono-to-Binaural may not be fair. As the data distribution for the two datasets is different, the baseline methods may not generalize well. The task of mono-to-binaural is very specific to the microphone array configuration and room information. A few shot settings can be tried for baseline methods to conduct a more thorough comparison.
- There exists a discrepancy between subjective and objective results on Binaural Speech for the proposed method. This paper does not go into depth and explains why the discrepancy happens. More explanations and analysis of why objective metrics do not model perceptual quality would be good.
- To verify the generalization ability of the proposed method, experiments on additional datasets are needed. Spatial sounds from DCASE 2016 Task 2 are simulated. There are also real spatial sound recordings from the DCASE community these years, such as DCASE 2022 task 3. Furthermore, how does the proposed method generalize speech with moving sources?

**Questions:**

N/A

---

### Official Review · Reviewer_xXge · 2024-11-04

**Soundness:** 1
**Presentation:** 2
**Contribution:** 2
**Rating:** 3
**Confidence:** 4

**Summary:**

This paper explores position-conditioned mono-to-binaural speech synthesis. It first constructs a new benchmark, TUT, for this task to test different models. Moreover, it proposes a new method, BinauralZero, which combines the parameter-free methods (i.e., geometric time warping and amplitude scaling) and the diffusion-based vocoder pre-trained on single-channel waveform. Given both objective and subjective evaluation, BinauralZero claims subjectively matched or stronger quality than existing neural-based mono-to-binaural models.

**Strengths:**

For mono-to-binaural speech synthesis, this paper proposes a new method and gives a well summary of related works. Moreover, both subjective and objective measurements are adopted to evaluate different methods.

**Weaknesses:**

1. Evaluation methods: The subjective evaluation in this work is not convincing to me. Firstly, as a test dataset, the samples in TUT are too short. This work explores position-conditioned mono-to-binaural speech synthesis. However, the samples in proposed TUT have a length of 2 seconds, which are not enough to demonstrate the change of position. Moreover, when visiting the test samples of Binaural Speech, I find the samples are segmented with a length of 6 seconds which are also shorter than previous methods such as BinauralGrad. A large amount Binaural Speech test samples in this work cannot show the variation of position. I think longer samples should be included in subjective evaluation, which can clearly show the position-conditioned synthesis quality.

2. Human evaluation: Why the DSP methods are not compared in Table 1? In previous works like WarpNet and BinauralGrad, DSP methods without trainable parameters have shown strong subjective quality. Moreover, in Appendix B, human evaluation details are introduced, where over 30 human raters are asked to give scores on 50 samples from each method. Why the 95% confidence intervals of the MOS scores are very high?

3. Novelty: The techniques including GTW, AS, and diffusion-based vocoder used in this work are not new. Each of them has already been explored in previous works or well-developed. This work combines these components, constructing a prior with training-free method and refining the prior with a pre-trained diffusion-based vocoder (training from scratch with 60k hours data in this work). The key advantage and innovation of proposed method are not very clear.

4. Efficiency: In training process, this method requires a vocoder pre-trained on 60k-hour LibriLight. It trains a WaveFiT vocoder from scratch and does not describe the time cost, e.g., GPU hours of this pre-training process. In inference process, it incorporates diffusion-based vocoder for waveform refinement, which will result in slower inference speed than DSP and WarpNet since each sampling step is calculated in the waveform space.

**Questions:**

1. Why most of the test samples cannot show the consistent change of position? Binaural speech synthesis is different from vocoder or text-to-speech synthesis. With a limited sample length (e.g., 2s), how to effectively demonstrate the quality of position-conditioned synthesis task?
2. Is this method a training-based one or a training-free one?
3. Why WaveFit is selected as the vocoder? What is the performance of other neural vocoder?
4. Why AS is critical for BinauralZero performance?
5. Could you add more details of the training resources and the inference speed of this method?

---

### Note · Authors · 2024-11-26

I have read and agree with the venue's withdrawal policy on behalf of myself and my co-authors.